# Sleep-Related Rhythmic Movement Disorder in Young Children with Down Syndrome: Prevalence and Clinical Features

**DOI:** 10.3390/brainsci11101326

**Published:** 2021-10-06

**Authors:** Ceren Kose, Izabelle Wood, Amy Gwyther, Susiksha Basnet, Chloe Gaskell, Paul Gringras, Heather Elphick, Hazel Evans, Catherine M. Hill

**Affiliations:** 1School of Clinical and Experimental Sciences, Faculty of Medicine, University of Southampton, Southampton SO16 6YD, UK; ck1n15@soton.ac.uk (C.K.); izabelle_wood@live.com (I.W.); a.gwyther@nhs.net (A.G.); sb7g17@soton.ac.uk (S.B.); crgaskell@dcotors.org (C.G.); 2Evelina Children’s Hospital, London SE1 7EH, UK; Paul.Gringras@gstt.nhs.uk; 3Sheffield Children’s NHS Foundation Trust, Sheffield S10 2TH, UK; Heather.Elphick@sch.nhs.uk; 4Southampton Children’s Hospital, Southampton SO16 6YD, UK; hazel.evans@uhs.nhs.uk

**Keywords:** rhythmic movement disorder, rhythmie du sommeil, jactatio capitis nocturna, Down syndrome, sleep, head banging, body rocking, actigraphy, videosomnography

## Abstract

Sleep-related Rhythmic Movement Disorder (RMD) affects around 1% of UK pre-school children. Little is known about RMD in Down syndrome (DS). We aimed to determine: (a) the prevalence of RMD in children with DS aged 1.5–8 years; (b) phenotypic and sleep quality differences between children with DS and RMD and sex- and age-matched DS controls; and (c) night-to-night variability in rhythmic movements (RMs). Parents who previously reported RMs from a DS research registry of 202 children were contacted. If clinical history suggested RMD, home videosomnography (3 nights) was used to confirm RMs and actigraphy (5 nights) was used to assess sleep quality. Phenotype was explored by demographic, strengths and difficulties, Q-CHAT-10/social communication and life events questionnaires. Eight children had confirmed RMD. Minimal and estimated maximal prevalence were 4.10% and 15.38%, respectively. Sleep efficiency was significantly lower in RMD-cases (69.1%) versus controls (85.2%), but there were no other phenotypic differences. There was considerable intra-individual night-to-night variability in RMs. In conclusion, RMD has a high prevalence in children with DS, varies from night to night and is associated with poor sleep quality but, in this small sample, no daytime phenotypic differences were found compared to controls. Children with DS should be screened for RMD, which is amenable to treatment.

## 1. Introduction

Sleep-related rhythmic movement disorder (RMD) is a rare sleep disorder [1] with a likely prevalence of around 1% in UK pre-school children established by our recent community-based study of 1464 infants and toddlers [2]. The International Classification of Sleep disorders (III) (ICSD III) defines RMD as ‘stereotyped, repetitive rhythmic motor behaviour’ in the frequency range 0.5–2 Hz [1], taking place during the transition between wakefulness and sleep as well as during periods of sleep [3]. Rhythmic movements (RMs) include body rocking, head banging or head rolling [4]. RMs are believed to often resolve in the second or third year of life [1]; however, longitudinal data are limited. RMD is hypothesised to be a conditioned, self-soothing behaviour that mimics maternal movements, heartbeat or respiratory rhythm experienced in utero [3]. Moreover, rocking in general (including non-sleep-related) may be a soothing behaviour in response to current life events; patterns of rocking were found in 47% of 144 institutionally reared Romanian children, aged between a few weeks and 43 months at the time of UK entry for adoption; this decreased to 18% by age 6 years [5]. However, the cause of RMD, and its impact on sleep and daytime function, is poorly understood [3].

Down syndrome (DS), or trisomy-21, has an incidence of 1 in 862 live births in England [6] and is the most common cause of learning disability in the developed world. Physical and developmental features of DS predispose these children to a number of sleep disorders, including obstructive sleep apnoea (OSA) [7], which is estimated to affect between 30 and 60% of the DS population [8]. In young children, behavioural insomnia is a likely contributor to short sleep duration [9]. Restless sleep is a common complaint in children with DS [10] but is poorly understood.

DS is often co-morbid with Autism Spectrum Disorder (ASD) and Attention-Deficit Hyperactivity Disorder (ADHD) and these disorders have independently been reported to be associated with RMD [3,11]. 

A diagnostic criterion for RMD is that it results in either significant sleep disturbance, physical injury or daytime effects. Despite this, there has been limited study of the impact of this disorder on sleep quality in the home environment. Sleep disturbance can have detrimental effects on cognitive function, daytime behaviour and quality of life in typically developing (TD) children as well as those with DS [12]. However, these effects may have more significant implications in children with DS due to their pre-existing medical impairments and limited cognitive reserves [13,14]. Furthermore, sleep may influence functional maturation of the brain in the early years of life, with evidence in toddlers with DS that sleep may facilitate language learning [15]. Early recognition and treatment of sleep disorders, therefore, may benefit these children more than their TD peers. 

We aimed to examine the prevalence of RMD in a sample of young children with DS and to identify behavioural, life-event and sleep quality differences between cases and age- and sex-matched controls with DS. Moreover, we considered intra- and inter-individual night-to-night variability of RMs in those with confirmed-RMD to explore which factors may influence the severity of RMD on different nights/in different participants. 

## 2. Materials and Methods

### 2.1. Ethics Approval

The study was approved by the UK National Research Ethics Committee (REC: 15/WA/0294) and the University of Southampton ethics committee (ERGO: 17342.A1). Written parental consent was obtained for all participants.

### 2.2. Participant Recruitment

Potential cases of RMD were identified from a research registry of 202 children, aged eighteen months to eight years, representative of a community population of children with DS, recruited between 2012 and 2014 for a study of OSA [16]. 

Potential cases were identified from responses to the following Child Sleep Habits questionnaire (CSHQ) [17] item: ‘Child falls asleep with rocking or rhythmic movements?’. Overall, 69/195 parents responding to this question answered ‘usually’ or ‘sometimes’. Efforts were made to contact these parents by telephone and e-mail to ask the follow-up question, “Does your child show repetitive actions such as body rocking or head banging while falling asleep, or during the night?”, and it was established whether the child had possible current-RMD or no RMD at all (i.e., parents may have misunderstood the question in the CSHQ). 

Those who answered ‘yes’ or ‘not sure’ were invited to participate in the study as potential RMD-cases, whereas those who answered ‘no’ were invited to participate as potential controls (unless they had a symptom description of resolved-RMD). Controls were matched from the database to each potential-RMD case based on gender and age.

Home visits were then arranged with all participating families.

### 2.3. Measures

#### 2.3.1. Questionnaires

The following questionnaires were completed by parents at the home visit:A demographic questionnaire comprising age, gender, ethnicity, parent occupation, highest parental education level attained, and geographical location based on postcode.The Child’s Sleep Habits questionnaire (CSHQ), consisting of 33 items, which can be subdivided into eight subscales: bedtime resistance, sleep onset delay, sleep duration, sleep anxiety, night wakings, parasomnias, sleep-disordered breathing, and daytime sleepiness. Items are rated on a 3-point scale according to frequency of occurrence from usually to rarely. It is based on common clinical symptom presentations of the most prevalent paediatric diagnoses according to the ICSD. Psychometric properties show satisfactory test–retest reliability for both normal and clinical populations [17]. The CSHQ has previously been used to assess sleep problems in children with DS [10,18]. A total score threshold of 41 and above detects clinical sleep problems with a sensitivity of 0.80 and specificity of 0.72.A life events questionnaire, with 21 items, which focuses on recent life events [19] to ascertain any negative life events (i.e., death of an immediate family member, parental divorce, etc.) that may impact the child’s behaviour. In a typical adult population, 7–8 events are reported.The Strengths and Difficulties Questionnaire (SDQ), a 25-item behavioural screening tool divided into 5 sub-scales: emotional symptoms, conduct problems, hyperactivity, peer problems and a pro-social scale. Psychometric properties indicate good reliability and validity [20]. The SDQ was previously used in children with DS [21]. Age-appropriate versions were used. A total difficulties score of 0–13 (aged 4 years and older) or 0–12 (aged 2–4 years) is considered normal, and 14–16 and 13–15, respectively, are considered borderline.The Quantitative Checklist for Autism in Toddlers (QCHAT-10), a 10-item screening instrument for ASD in children under four years old. The QCHAT demonstrates good psychometric properties and external validity in discriminating between autism and developmental delay in children [22], with a recommended clinical threshold score of 3 [23].The Social Communication Questionnaire (SCQ), a 40-item screening tool for ASD validated in children aged four years and above with a recommended clinical threshold score of 15 [24]. Psychometric studies support the assessment of ASD in children with DS using the SCQ [25].

#### 2.3.2. Home Videosomnography

Three nights of home video recording in the child’s bedroom enabled characterization of RMs in potential-RMD cases only. As per our previous prevalence study [2], an infrared motion-sensitive camera was used (Wansview NCM624W H.264 720P MegaPixel Indoor Wireless IP Camera).

#### 2.3.3. Actigraphy

Five consecutive nights of wristwatch actigraphy were recorded using the MicroMini-Motionlogger^®^ Actigraph (Ambulatory Monitoring Inc., New York, NY, USA). Actigraphy was not used as a diagnostic tool for RMD (where polysomnography remains the gold-standard), but to assess sleep duration and quality. Studies in TD children demonstrate that actigraphy can reliably determine total sleep time and sleep efficiency in concordance with polysomnography [26]. The watch was worn on the non-dominant wrist at bedtime and taken off at morning awakening. A sleep diary was used in conjunction with the actigraph; recording of lights-out enabled sleep onset latency to be computed. Sleep measures of interest in this study were:A.Sleep period = total number of minutes from sleep-onset to morning awakening time;B.Sleep minutes = total number of minutes scored as sleep during the sleep period—excludes any periods of wakefulness;C.Sleep efficiency (%) = proportion of minutes scored as sleep during the sleep period;D.Sleep onset latency = minutes taken from lights-out to sleep onset.

### 2.4. Data Analysis

All questionnaire data were entered into IBM SPSS v26 and scored where relevant to produce sub-scale or total scores. 

Videosomnography: Data were downloaded and visually inspected in real time using Windows Media player. As there is no agreed method of scoring RMs using this technology, a novel RMD video classification tool was used. RMs were scored for onset-time, duration and frequency (Hz), and semiology of the RMs was described. Discrete episodes were determined as having a minimum inter-movement duration of 10 s. A note was made of whether the child was awake/asleep or whether this was unclear, whether there was any vocalisation and whether the child’s body struck an object (e.g., wall/headboard) during the episodes. The inter-rater reliability of this tool was explored; the Kappa measure of agreement for type of movement was high (Kappa = 1.0, *p* < 0.001), whereas the Kappa measure of agreement for whether the child was awake or asleep was low (Kappa = 0.09, *p* = 0.63). Movements that were suspected of meeting RMD diagnostic criteria were confirmed by a European Sleep Research Society-certified somnologist (CMH), using the diagnostic criteria in the ICSD III [1].

Actigraphy: Actigraphic data were analysed using Action W2.7 software with the Sadeh algorithm validated for use with children [27]. First, raw data were visually inspected to reject any epochs/nights where the actigraph had been removed (e.g., taken off by child in the middle of the night). Next, using the sleep/wake times reported in the sleep diary, the limits for each night were set. Using the automated algorithm, mean values for sleep period, sleep minutes, sleep efficiency and sleep onset latency across the five nights of data capture were calculated.

### 2.5. Statistical Analysis

Statistical analysis was performed using IBM SPSS v26. Appropriate statistical tests were used depending on the distribution of data; normality was explored using Shapiro–Wilk tests for smaller samples. Categorical differences were explored using the Pearson’s Chi-Squared test and group differences explored using an independent-samples *T*-test or a Mann–Whitney-*U* test, depending on the data distribution.

## 3. Results

### 3.1. Prevalence

Of the 69 parents who answered ‘usually or sometimes’ to the CSHQ RM screening question, 29 were able to confirm that their child did not have—and never had—RMs, suggesting this question was poorly understood. It was not possible to contact 22 families. A further 18 parents confirmed that their child had likely RMs associated with sleep and participated in further study. Detailed home assessment confirmed RMD in only 8/18 of these children. These eight children were sex- and age-matched to controls who did not have—and never had—RMs (Figure 1).

The prevalence of RMD was estimated in three ways to allow for missing data as per our previous prevalence study [2]. All three prevalence figures can be reviewed in Figure 2.
1.Maximal prevalence

This was calculated by excluding those who answered no during the follow up call and those subsequently shown by objective measures not to have RMD. It was assumed that all those who could not be confirmed were positive cases.
$$\mathrm{MAX}=\frac{\begin{array}{c} \mathrm{Total who screened yes}-\\ (\mathrm{total who answered no at follow}-\mathrm{up call}+\\ \mathrm{confirmed to not have RMD}) \end{array}}{\mathrm{Total}\;\mathrm{who}\;\mathrm{answered}\;\mathrm{the}\;\mathrm{screening}\;\mathrm{question}}\times 100\;=\frac{69-\left(29+10\right)}{195}\times 100\;=15.38\%\;\left[95\%\;\mathrm{CI}\;\left(10.32,\;20.44\right)\right]$$
2.Likely prevalence

The likely prevalence was calculated as all objectively confirmed cases plus all potential cases (cases where confirmation was not possible, i.e., those who answered yes to the initial screening question and for whom the follow-up call failed to exclude the presence of RMs) multiplied by the ratio of parentally reported cases that were objectively confirmed at home visit. This accounted for the families who were unable to be contacted or unwilling to participate in the study.
$$\mathrm{LIKELY}=\frac{\begin{array}{c} \mathrm{Objectively confirmed cases}+\\ \left(\mathrm{Potential cases}\times \frac{\mathrm{Objectively}\;\mathrm{confirmed}\;\mathrm{cases}\;}{\mathrm{Total}\;\mathrm{number}\;\mathrm{of}\;\mathrm{home}\;\mathrm{visits}}\right) \end{array}}{\mathrm{Total}\;\mathrm{who}\;\mathrm{answered}\;\mathrm{the}\;\mathrm{screening}\;\mathrm{question}}\times 100\;=\frac{8+\left(22\times \frac{8\;}{18}\right)}{195}\times 100\;=9.12\%\;\left[95\%\;\mathrm{CI}\;\left(5.08,\;13.16\right)\right]$$
3.Minimal prevalence

The minimal prevalence was calculated as only confirmed cases divided by the total number of those whose parents answered the initial screening question. It was assumed that none of those who could not be contacted had RMD.
$$\mathrm{MIN}=\frac{\mathrm{Objectively}\;\mathrm{confirmed}\;\mathrm{cases}}{\mathrm{Total}\;\mathrm{who}\;\mathrm{answered}\;\mathrm{the}\;\mathrm{screening}\;\mathrm{question}}\times 100\;=\frac{8}{195}\times 100\;=4.10\%\;\left[95\%\;\mathrm{CI}\;\left(1.32,\;6.88\right)\right]$$

### 3.2. Demographics

There were no statistically significant differences between cases (*n* = 8) and controls (*n* = 8) for any of the demographic measures (Table 1), including between any questionnaire subscales. All children had CSHQ scores above the recommended clinical cut off of 41 [17], indicating a high prevalence of sleep problems irrespective of RMD. Relatively few life events were reported (typical adult samples report 7–8) and no parents reported more than 7. Total scores on the SDQ indicated that the groups did not have behavioural difficulties, although four children in each group had borderline total scores. Two children in the control group had Q CHAT-10 scores above the screening threshold for ASD but no children scored positively in either group using the social communication questionnaire.

### 3.3. Actigraphy

As illustrated in Figure 3, sleep efficiency was significantly lower in children with RMD compared to controls: 69.1% (SD = 11.87) vs. 85.2% (SD = 7.89), *p* = 0.013. No statistical significances were found for sleep minutes (460.65 min (SD = 32.73) vs. 460.91 min (SD = 103.05), *p* = 0.99), sleep period (694.25 min (SD = 34.74) vs. 666.16 min (SD = 33.3), *p* = 0.12), or sleep onset latency (53.00 min (SD = 12.50) vs. 55.00 min (SD = 24.00), *p* = 0.89).

### 3.4. Video Scoring

Table 2 outlines the characteristics of the RMD-cases.

Figure 4A illustrates the total number of RM clusters during the time in bed, summed across all participants and across all three nights of recording. It is notable that RMs predominantly occurred at the beginning of the night, prior to sleep onset and in the second half of the night. Figure 4B shows the total number of RM episodes summed across all participants separated by each consecutive night of recording in time intervals of 30-min. Plotting each of the three nights individually illustrates the night-to-night variability across all participants.

To explore night-to-night variability further, the total number of episodes across all cases per night of recording was calculated. Night 1 had fewer total episodes (251) than night 2 (356). The intra-individual night-to-night variability is illustrated in the duration distribution plot in Figure 5. For some participants, the total duration of RMs differed significantly from night to night (i.e., case E), whereas in others it remained quite stable (i.e., case D). There was also a wide inter-individual variability in duration of episodes. The range of time spent rocking during time in bed across all eight cases ranged from 0.47 to 104.30 min.

## 4. Discussion

To our knowledge, this is the first study to explore the prevalence and characteristics of RMD in children with DS. We found that there was a significantly higher prevalence of RMD in this population of children than in the general population, with a likely prevalence of 9.12% compared to a likely prevalence of 0.96% in a community sample of infants and toddlers in the UK [2]. It should, however, be noted that the average age of the population of children with RMD in the community study [2] was younger than that of the DS children reported here. RMs are thought to commonly resolve in the second or third year of life [1], with persistence at age five years only occurring in about 5% of cases [1]. The children in our study with persistent-RMD were, on average, 4.56 years old. This indicates that children with DS are also more likely to experience persistent-RMD compared to TD children. This is also supported by the fact that there were no differences in age between the eight RMD cases confirmed in this study and the 61 other children whose parents responded positively to the screening questionnaire at baseline. There is a lack of research to explain why RMD persists in some children and why spontaneous resolution occurs in others. Prospective cohort studies of children with RMD (involving both TD children and children with neurodevelopmental disorders) could help bridge this gap in the literature.

Our data suggest that RMs predominantly occur at the beginning and the middle of the night. This is in line with normal sleep architecture with the first half of the night being dominated by NREM (non-rapid eye movement) stage 3 (or deep sleep), while the second half of the night is dominated by REM (rapid eye movement) and NREM stage 2 light sleep. Brief arousals occur at the end of each sleep cycle, generally followed by an immediate return to sleep. In RMD, the child may use RMs to self-soothe both when they are trying to fall asleep at the beginning of the night and again following physiological arousal from lighter sleep as the night progresses (i.e., during sleep cycle transitions).

A possible explanation for the higher prevalence of RMD in children with DS is that they have greater difficulty settling to sleep, and thus, use RMs as a conditioned soothing behaviour—perhaps to mimic the soothing they would typically seek from their parents before falling asleep. Vrugt et al. investigated how vertical rocking (at frequencies of 0, 0.5, 1 and 1.5 Hz) affected the arousal level of 64 infants aged two-months [28]. They found that arousal decreased with increasing frequency of rocking and that more infants slept and fewer cried when rocked at the highest tested frequency of 1.5 Hz [28]. In support of this hypothesis, we previously reported that parents of infants and toddlers with DS were more likely to engage in effortful soothing at bedtime to settle their child to sleep compared to parents of TD peers [9].

The higher prevalence of RMD in children with DS may be related to the neurocognitive impairments and developmental delay in this population of children. While links between RMD and DS have not previously been made in the literature, links between RMD and other neurodevelopmental disorders (i.e., ASD and ADHD [3,11]) have. These conditions are known to be more common in children with DS [9], providing a potential indirect link between RMD and DS. Nonetheless, we did not find any significant phenotypic differences between RMD-cases and controls, nor any difference in scores from screening questionnaires for ASD or behavioural disturbance. Further larger studies in children with neurodevelopmental disorders may potentially shed further light on the aetiology of RMD.

RMD has been reported in association with sleep-disordered breathing. Chirakalwasan and colleagues reported a single case study of a 38-year-old male in whom a dramatic reduction in RMD symptoms occurred following the treatment of their OSA [29]. Lending support to this, Groswasser et al. reported a reduction in obstructive breathing events when sleeping on a rocking mattress in 16 out of 18 infants studied with OSA [30]. It is unclear how similar these rocking movements were to the movements that are characteristic of RMD. However, it is plausible that children with DS, who commonly have OSA [16], may rock to decrease obstructive breathing events, although this would not explain RMs at sleep onset or after arousals from sleep. Sleep-disordered breathing was not measured within this particular study to confirm or refute this theory, but clinical experience of the supervising author (CMH) indicates that OSA is not a common co-morbidity in children with RMD.

A key finding of this study was the significantly worse sleep quality measured objectively in children with RMD compared to controls (69.1% vs. 85.2%). This has not been previously documented using objective measures in a case-control study design in children with RMD. Children with DS are known to have restless sleep [10] but our data suggests that this is significantly worsened by RMs. Sleep efficiency in our sample is a measure of time asleep rather than time in bed, and the low levels observed are likely to be clinically significant. In support of this, an actigraphic study of 29 toddlers with DS and 24 TD toddlers divided children into poor sleepers (sleep efficiency < 80%) and good sleepers (>80%). It was found that 66% of the children with DS had poor sleep compared to 15% of the TD children. Across the groups, parents reported greater deficits in language (both vocabulary and syntax) in toddlers with poor sleep efficiency [15].

While we failed to find any systematic differences in any other objective sleep parameters across the sample, there were some notable findings. Sleep onset latency was prolonged (53 min case and 55 min controls) compared to normal values of <30 min [31]. Sleep onset latency at this age may be extended by child behaviours such as bedtime resistance and bedtime fears, as well as physical discomfort or inappropriately early bedtimes. Furthermore, total sleep time was short, averaging a little over 6.5 h compared to expected sleep durations at this age of greater than 9 h [32]. We did not measure day naps, which may compensate for short night sleep durations; however, our prior study in infants and toddlers with Down syndrome suggested that longer day naps did not compensate for short night sleep [9].

We found that there was considerable intra-individual night-to-night variability in RMs. The total number of episodes across all cases on night 1 were fewer (251) than on night 2 (356). It is possible this may have been a result of a first night effect with RMs suppressed because the child was aware of the camera; however, this does not provide an explanation for night 3 having the least number of RM episodes (143). The findings suggest that objective measurement of disorder severity should include multiple nights, e.g., to eliminate the influence of the first-night effect on results. Moreover, factors driving night-to-night variability may shed light on the nature and severity of RMD. For example, was the increase in episodes on the second night only due to the first-night effect, or could daytime experiences trigger more/fewer episodes on certain nights, as is well described in parasomnias? Furthermore, the degree of inter-individual variability in RM severity was striking in this study. While other episodic sleep disorders, such as OSA, are assessed using detailed severity criteria [33], there is no standardised approach for determining the severity of RMD. This is an important area for future guideline development to inform treatment decisions and monitor treatment outcomes. For example, a child with severe/persistent RMD may be directed towards active treatments, whereas a child with mild RMD may need reassurance.

We performed home videosomnography for three nights for each child where parents reported likely RMD symptoms, but only confirmed RMs in 44.4% of cases. This is consistent with our population prevalence study where parental report was not confirmed by three nights of objective home videosomnography in 46.7% of cases [2]. While three nights of recording may have not been enough to capture the child’s usual sleep habits, it is also possible that parental report is unreliable. It is unclear what parents were reporting when they offered positive responses to both the CSHQ screening questionnaire and our further probe question. Future qualitative approaches such as ‘think aloud’ cognitive interviews with parents may shed light on their interpretation of this item. However, our data suggest that previous population estimates based on parental report alone (without any objective confirmation) may significantly overestimate prevalence.

## 5. Limitations

There are a number of limitations to our data. Firstly, with respect to prevalence data, there is a risk of bias in our sampling frame. The children were recruited from a research registry rather than directly from a community of children with DS. The original research registry sample, however, were not recruited based on RM history; rather, this was an incidental finding. Moreover, multiple recruitment routes were used to increase the generalisability of the results, including local children’s hospitals, DS parent support networks and parent word-of-mouth [16]. Secondly, we only sampled from the 69 families who responded positively to the CSHQ screening question ‘Child falls asleep with rocking or rhythmic movements?’. This question had low specificity to detect RMD but we cannot be confident of its sensitivity without also studying children of parents who responded ‘rarely or never’ to this question. However, based on our clinical understanding of RMD, it would seem likely that the question would be sensitive to detect children with RMD at sleep onset, although may miss children with RMs only after they have fallen asleep. Importantly, the aim of this research was to assess the prevalence of RMD. Any missed cases would have increased this prevalence and, therefore, strengthened our findings. Thirdly, we did not have sufficient resources to conduct videosomnography in controls. We cannot, therefore, be certain that these children did not have RMD, although clinical sense predicts a high sensitivity for our supplementary screening question. Furthermore, actigraphic differences between cases and controls offers support to this. Finally, the number of identified cases was small, but the findings were strengthened by using a case-control design. The small sample size may have obscured subtle phenotypic differences between the cases and controls. We explored this further by removing the three milder cases (B, C and D—see Figure 4B) and their controls from the analysis. This did not reveal any significant phenotypic differences between cases and controls other than the difference in sleep efficiency, which remained significant. Future studies of larger samples of children may shed further light on associated behavioural and developmental characteristics. Nonetheless, this reinforces the significance of the positive finding of poorer sleep quality in children with RMD.

Our data included five nights of actigraphic recording. Acebo et al. [34] found that “5 or more nights of usable recordings are required to obtain reliable actigraph measures of sleep for children and adolescents”. However, they also found that “measures of sleep minutes and sleep period were less reliable and may require 7 or more nights for estimates of stable individual differences” [34]. There are limited published actigraphic data in this population, but Esbensen and colleagues studied 30 school-aged children with DS, using the same Ambulatory Monitoring Inc. devices as used in our study over seven consecutive nights [35]. Interestingly, the mean sleep efficiency data across their sample (mean: 87.4%; SD: 6.04) were very similar to those of the control children in our sample (mean: 85.2%; SD: 7.89), providing some confidence in our data. In the future, studies including longer periods of actigraphy recording could improve the validity of results.

A recent study on novel automatic 3D-video analysis [36] suggests that it may be an alternative approach to the quantification of RMs, offering a wider angle of view and more efficient data processing compared to time-consuming manual 2D video analysis. Computerised approaches can dramatically increase the speed of video scoring and have been found to match the results of manual scoring to a high degree [36]. While the inter-rater reliability for our novel RMD video classification tool had a high measure of agreement for type of movement, there was poor agreement for awake/asleep classification. This parameter was difficult to assess when the child was facing away from the camera or when engaged in RMs with eyes closed. Whether a child is awake or asleep during RMs is relevant to treatment approaches. For example, behavioural approaches are unlikely to be effective if RMs arise from sleep. Polysomnography, the gold standard for objective sleep assessment [37], offers this information and would have been a useful diagnostic adjunct in this study to distinguish RMs from other high amplitude movements. However, many children inhibit RMs during polysomnography studies [36]; even so, this information may helpfully guide treatment approaches as a child who cannot suppress movements occurring due to waking in a laboratory setting may be more resistant to treatment approaches and vice versa.

## 6. Conclusions

This study has generated a number of novel findings. Firstly, using stringent objective diagnostic criteria, RMD is significantly more common in children with DS than it is in a general population of infants and toddlers [2]. Secondly, RMD is associated with significantly worse sleep quality. Intra-individual night-to-night variability in the severity of episodes indicates that RMs should be measured over multiple nights to obtain a true reflection of sleep disruption in children with RMD. Furthermore, the significant inter-individual variation highlights the need for a robust severity rating score for this condition as for similar episodic sleep disorders.

Finally, children with DS have pre-existing neurocognitive impairments and limited cognitive reserves to compensate for the reductions in sleep quality observed in this study. RMD is amenable to treatment, and children with DS should be screened and actively treated to promote optimal healthy sleep and daytime wellbeing.

## Figures and Tables

**Figure 1 brainsci-11-01326-f001:**
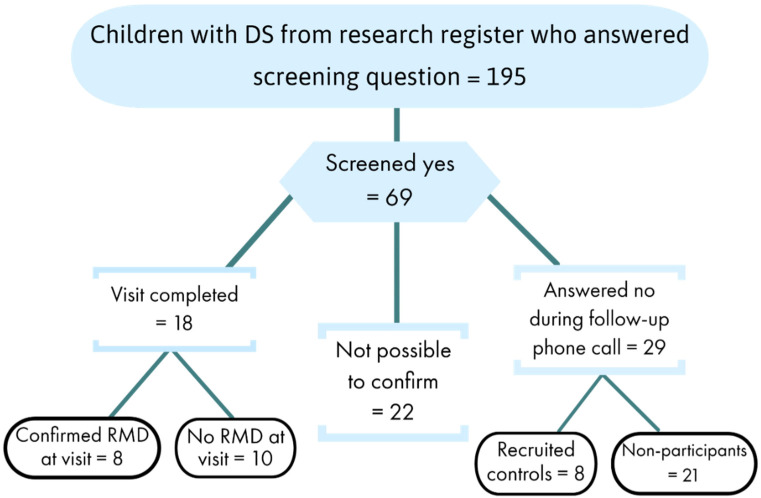
Flow diagram showing outcome of all 69 potential Rhythmic Movement Disorder (RMD)-cases.

**Figure 2 brainsci-11-01326-f002:**
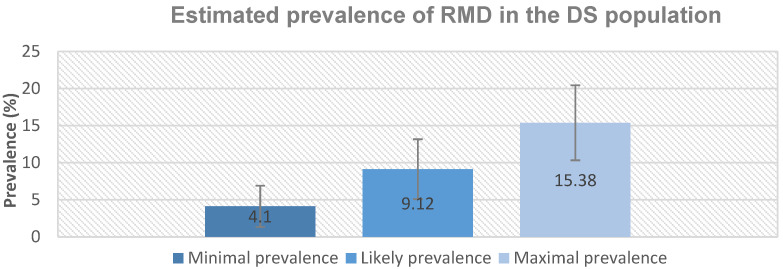
Prevalence estimates of Rhythmic Movement Disorder. Error bars indicate 95% confidence intervals.

**Figure 3 brainsci-11-01326-f003:**
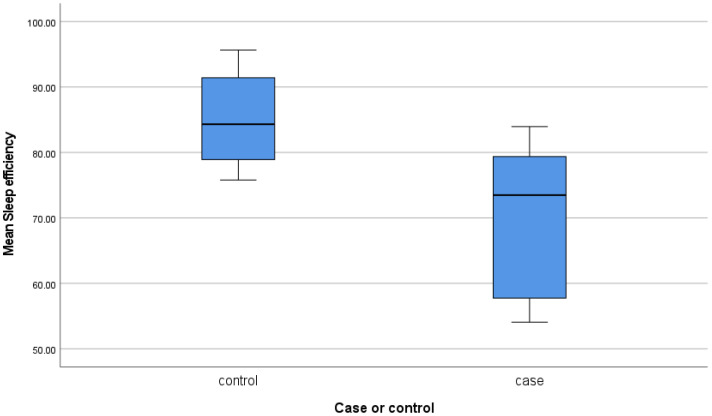
Boxplot showing sleep efficiency of children with and without RMD.

**Figure 4 brainsci-11-01326-f004:**
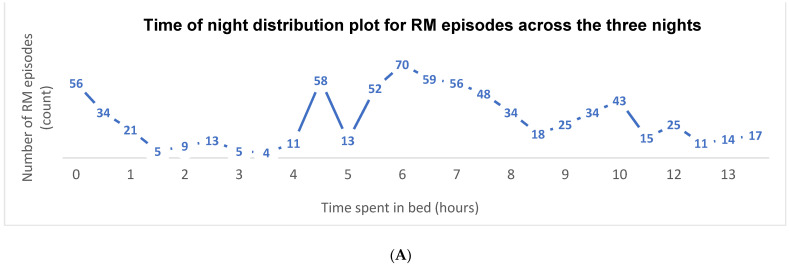
(**A**) Time of night distribution plot for rhythmic movement episodes across the three nights. (**B**) Rhythmic movement time of night distribution plot for each of the three nights.

**Figure 5 brainsci-11-01326-f005:**
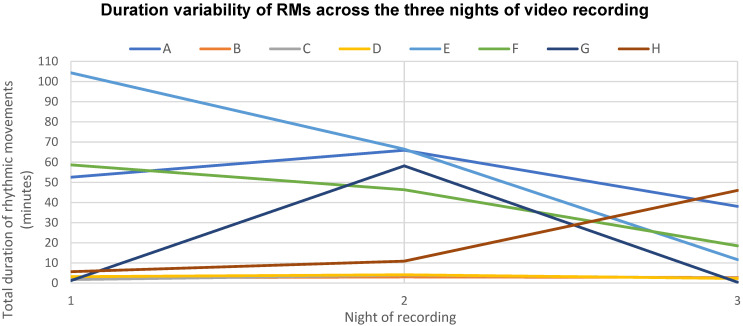
Duration distribution plot illustrating night-to-night variability, across all three nights of video recording, with each child represented individually (cases A–H). While some children had a consistent pattern of duration of rhythmic movements across the three nights, others displayed a high intra-individual night-to-night variability.

**Table 1 brainsci-11-01326-t001:** Demographic and questionnaire data. CSHQ = Child Sleep Habits Questionnaire; SDQ = Strengths and Difficulties questionnaire; SCQ = Social Communication Questionnaire; QCHAT = Quantitative Checklist for Autism in Toddlers.

Variable	Cases *n* = 8 Mean (SD)		Controls *n* = 8 Mean (SD)		Significance Level
Age in months	54.75 (21.24)		56.00 (20.68)		*p* = 0.91
Gender	4M:4F		4M:4F		*p* = 1.00
Ethnicity	White British	*n* = 6	White British	*n* = 7	*p* = 0.58
	Black British	*n* = 1	Black British	*n* = 1	
	Asian British	*n* = 1	Asian British	*n* = 0	
Parent educational level	N/A	*n* = 1	N/A	*n* = 0	*p* = 0.21
	GCSE *	*n* = 0	GCSE *	*n* = 1	
	A-Level	*n* = 2	A-Level	*n* = 5	
	Degree or higher	*n* = 5	Degree or higher	*n* = 2	
CSHQ total sleep disturbance score	53.88 (5.44)		54.50 (7.76)		*p* = 0.86
Total Recent Life Events Score	2.13 (1.81)		2.88 (2.30)		*p* = 0.48
Total SDQ Difficulties Score	11.75 (4.59)		13.63 (2.07)		*p* = 0.31
Total QCHAT-10 score **	0.50 (0.71)		4.00 (2.65)		*p* = 0.18
Total SCQ score ***	9.33 (3.08)		8.20 (3.03)		*p* = 0.56

* GCSE = General Certificate of Secondary Education. ** cases: *n* = 2; controls: *n* = 3. *** cases: *n* = 6; controls: *n* = 5.

**Table 2 brainsci-11-01326-t002:** Rhythmic movement characteristics for each RMD-case participant seen in video scoring.

Participant	Age (months)	Gender	Number of Episodes per Night (Mean, (Range))	Duration of Episodes in Seconds (Mean, (Range))	Total Time Spent in RMs in Minutes per Night (Mean, (Range))	Semiology			
						Movement Types	Range of Frequencies of Rhythmic Movements (Hz)	Impact against Object (i.e., Mattress/Cushioning Apparatus)	Vocalisation
A	54	F	48.0, (39–58)	65.2, (1–723)	52.2, (38.1–65.9)	Head banging, body rocking	0.20–1.17	Yes	Yes
B	18	F	12.3, (5–19)	14.0, (3–98)	2.8, (2.6–3.1)	Leg banging, hand banging, arm waving	0.30–2.00	Yes	No
C	66	F	7.3, (5–9)	27.1, (4–77)	3.2, (2.2–3.9)	Rubbing against hands/toy	0.36–0.81	No	No
D	30	M	11.7, (9–13)	14.6, (3–55)	2.8, (2.1–3.5)	Head banging and rolling, body rocking, leg banging and rolling, arm banging, hand banging	0.31–2.00	Yes	No
E	75	M	71.3, (36–94)	51.1, (2–388)	60.8, (11.7–104.3)	Head banging, body rocking, bouncing	0.26–1.50	Yes	No
F	74	M	59.7, (31–85)	41.4, (2–263)	41.2, (18.5–58.7)	Head-banging	0.60–2.50	Yes	No
G	50	F	44.7, (5–121)	26.8, (3–239)	20.0, (0.5–58.1)	Lateral hip movements	0.31–1.00	No	No
H	71	M	8.0, (3–13)	156.5, (4–849)	20.9, (5.7–46.0)	Body rolling	0.59–1.53	No	No

## Data Availability

The data presented in this study are available on request from the corresponding author.

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
