# Peer review of "Sleep-Related Rhythmic Movement Disorder in Young Children with Down Syndrome: Prevalence and Clinical Features"

_brainsci, 2021, doi:10.3390/brainsci11101326_

Round 1
Reviewer 1 Report
The paper is accurate and well written. The design of the study is clear, the results well presented and several limitations disclosed. The paper illustrates the prevalence and clinical features of sleep-related Rhythmic Movement Disorder (SRRMD) in children with Down Syndrome, assessed by means of questionnaires and confirmed by video-recording. However, I would recommend more caution with sleep data assessed by actigraphic recording. It is true that a good concordance between actigraphy and polysomnography has been demonstrated but, to my knowledge, there are no validated studies in children with SRRMD. Polysomnography remains the gold standard to examine sleep features in these patients, given the possible occurrence of rhythmic movements during periods of relaxed wakefulness, sleep transitions or different NREM sleep stages, with different implications on sleep efficiency. I would suggest to underline these aspects and to highlight in the limitations section the lack of a polysomnographic assessment. Would the authors consider in future studies to confirm their sleep data using a PSG study? I have no further comments.
Author Response
Reviewer's comment:
The paper illustrates the prevalence and clinical features of sleep-related Rhythmic Movement Disorder (SRRMD) in children with Down Syndrome, assessed by means of questionnaires and confirmed by video-recording. However, I would recommend more caution with sleep data assessed by actigraphic recording. It is true that a good concordance between actigraphy and polysomnography has been demonstrated but, to my knowledge, there are no validated studies in children with SRRMD. Polysomnography remains the gold standard to examine sleep features in these patients, given the possible occurrence of rhythmic movements during periods of relaxed wakefulness, sleep transitions or different NREM sleep stages, with different implications on sleep efficiency. I would suggest to underline these aspects and to highlight in the limitations section the lack of a polysomnographic assessment. Would the authors consider in future studies to confirm their sleep data using a PSG study? I have no further comments.
Response:
We would like to thank the reviewer for their positive comments on the paper. With respect to their thoughts on actigraphy, we agree that actigraphy is not a diagnostic tool for RMD. It is not used in clinical practice or in this paper to diagnose RMD – it was used however, to assess sleep variables such as sleep duration and sleep efficiency. We have now added in a sentence in the manuscript to make this clearer. Please see page 3 line 140, “Actigraphy was not used as a diagnostic tool for RMD (where polysomnography remains gold-standard), but to assess sleep duration and quality”.
Also, please see the last paragraph in the limitations section (page 12 line 437), where it is outlined that PSG could have been a useful tool in the study. “…Whether a child is awake or asleep during RMs is relevant to treatment approaches. For example, behavioural approaches are unlikely to be effective if RMs arise from sleep. Polysomnography, the gold standard for objective sleep assessment [37] offers this information and would have been a useful diagnostic adjunct in this study to distinguish RMs from other high amplitude movements. However, many children inhibit RMs during polysomnography studies [36]; even so, this information may helpfully guide treatment approaches as a child who cannot suppress movements occurring from wake in a laboratory setting may be more resistant to treatment approaches and vice versa…”.
I hope that the changes made provide appropriate clarity.
Reviewer 2 Report
This paper describes the prevalence and effects on sleep quality of Rhythmic Movement Disorder (RMD) in children with Down syndrome (DS). The study found that the prevalence of RMD was higher in children with DS compared to that reported in the literature for typically developing children. In a small group of 8 children with DS and matched controls those with RMD had lower sleep efficiency. Children with DS are at high risk for sleep disorders and as disruption affects daytime functioning it is important that sleep problems are identified and treated to improve quality of life for the children and their families. Although the sample size was small, this is a rare condition, the data are novel and have important clinical relevance to children with DS and their families.
Major comments:
Figure 1 may be misleading as how many children had more than 9 hours sleep – the results report an average of 7.6 hours of sleep but the figure goes to 14 hours. The authors should consider presenting the data as percentage of episodes rather than the actual number per hour in bed. Further, were the higher rates 4-5 hours after going to bed associated with wake periods identified on actigraphy and were the higher rates after 9 hours associated with being awake in the morning?
Table 1 The authors could consider comparing the subscales of the various questionnaires as well as the total scores. Were there differences between groups for example in the sleep breathing disorders subscale of the CSHQ?
Were there any differences in the use of dummies, security blankets and other sleep transitional objects between groups?
Minor comments:
Was the actigraphy conducted during the week or on weekends as studies have shown that sleep patterns differ across the week.
Was wake after sleep onset able to be measured?
Page 4 line 167 ESRS needs to be spelled out.
Was the age different between the children included in the study and those that were not included? If the non-included children were older this would support findings in typically developing children that children may grow out of RMD.
Table 1 GCSC needs to be spelled out for those readers not familiar with these UK exams. Usually parental education is reported as high school and university undergraduate and postgraduate degrees.
Table 2. Suggest removing the 2 decimal places for number of episodes and duration as the data cannot be measured to this level of accuracy.
Author Response
[reviewer's comments in bold]:
This paper describes the prevalence and effects on sleep quality of Rhythmic Movement Disorder (RMD) in children with Down syndrome (DS). The study found that the prevalence of RMD was higher in children with DS compared to that reported in the literature for typically developing children. In a small group of 8 children with DS and matched controls those with RMD had lower sleep efficiency. Children with DS are at high risk for sleep disorders and as disruption affects daytime functioning it is important that sleep problems are identified and treated to improve quality of life for the children and their families. Although the sample size was small, this is a rare condition, the data are novel and have important clinical relevance to children with DS and their families.
We would like to thank the reviewer for their careful review of the paper and for their positive comments. Our responses to each of the comments are outlined in red below:
Major comments:
1. Figure 1 may be misleading as how many children had more than 9 hours sleep – the results report an average of 7.6 hours of sleep but the figure goes to 14 hours. The authors should consider presenting the data as percentage of episodes rather than the actual number per hour in bed. Further, were the higher rates 4-5 hours after going to bed associated with wake periods identified on actigraphy and were the higher rates after 9 hours associated with being awake in the morning?
-
- I presume the reviewer meant figure 4, and that there is a misunderstanding on the reviewer’s part where the total time in bed has been confused with the actigraphic measure of sleep minutes (total number of minutes scored as sleep – excluding any periods of wakefulness, which does have an average of 7.6 hours of sleep). The average sleep period (not excluding periods of wakefulness) was in fact 11.6 hours. Apologies that this was not clear but Figure 4 is related to time in bed, not total sleep minutes. Hence, the figure goes up to 13.5 hours as this was the maximum range of hours spent in bed.
- We have tried out the interesting idea of plotting as percentages rather than frequency of episodes, which predictably gives us a similarly-shaped graph. However, we feel on balance that the total number gives the reader a better impression of the sheer number of episodes that can occur in RMD. We are happy to defer to editorial judgement on this.
- We have expanded the size of the graphs to make the axes clearer to read and hopefully easier to interpret when RMs tend to occur (figure 4A) and the night-to-night variability (figure 4B).
- Further, we were able to measure any gross motor activity with actigraphy - including during times of higher rates of episodes in accordance with the video footage. However, a limitation of actigraphy was that it is not a reliable tool to distinguish RMs from other high amplitude movements associated with being awake. For this reason, we cannot confidently say that actigraphy showed us that the higher rates were associated with wake periods. Video scoring also had similar limitations, i.e. if the child was awake but with their eyes closed or if they were facing away from the camera. Our thoughts on the timing of episodes are outlined in the discussion as follows, “Our data suggests that RMs predominantly occur at the beginning and the middle of the night. This is in line with normal sleep architecture with the first half of the night dominated by NREM (non-rapid eye movement) stage 3 (or deep sleep) while the second half of the night is dominated by REM (rapid eye movement) and NREM stage 2 light sleep. Brief arousals occur at the end of each sleep cycle, generally followed by an immediate return to sleep. In RMD, the child may use RMs to self-soothe both when they are trying to fall asleep at the beginning of the night and again following physiological arousal from lighter sleep as the night progresses (i.e. during sleep cycle transitions).” Polysomnographic studies would have been able to confirm whether children were asleep/awake; the lack of the use of gold-standard polysomnography is highlighted in the limitations of this study.
2. Table 1 The authors could consider comparing the subscales of the various questionnaires as well as the total scores. Were there differences between groups for example in the sleep breathing disorders subscale of the CSHQ?
Thank you for this sensible suggestion, we have explored the subscales and found that there are no significant differences. We have added this information into the results section of the paper (page 6, line 233).
3. Were there any differences in the use of dummies, security blankets and other sleep transitional objects between groups?
There were minimal differences in the use of dummies, security blankets and other sleep transitional objects between groups. In such a small sample, the low numbers of users limited useful interpretation so this has not been included.
Minor comments:
1. Was the actigraphy conducted during the week or on weekends as studies have shown that sleep patterns differ across the week.
We didn’t control for weekday/weekend differences so actigraphy was conducted at any suitable time of the week for the researchers and the families. It is likely that any differences were counterbalanced between groups. The key finding in actigraphy was the difference in sleep efficiency between groups and we were not looking at circadian timing of sleep. Generally, we don’t see as much weekend/weekday differences in this age group as we see in i.e. adolescents (oldest child in this study was 8 years old). Inspection of the sleep diaries does not suggest noteworthy difference in bedtimes or wake up times between any of the days. However, this is an important point to keep in mind for any future studies – especially in older cohorts of children who may not have such consistent bedtimes.
2. Was wake after sleep onset able to be measured?
Please read the definitions of the actigraphic sleep measures used in our study below:
-
-
- Sleep period = total number of minutes from sleep-onset to morning awakening time
- Sleep minutes = total number of minutes scored as sleep during the sleep period – excludes any periods of wakefulness
- Sleep efficiency (%) = proportion of minutes scored as sleep during the sleep period
- Sleep onset latency = minutes taken from lights-out to sleep onset
-
We measured sleep period and sleep minutes so wake minutes can be calculated by subtracting the sleep minutes from the sleep period. We decided not to include this measure as we felt that sleep efficiency captured this well. We could go back to the source data to calculate wake minutes but we feel that it wouldn’t add much to the paper. Furthermore, wake after sleep onset is acknowledged to be an unreliable actigraphic measure (Meltzer LJ et al. Use of actigraphy for assessment in pediatric sleep research. Sleep Med Rev. 2012 Oct;16(5):463-75)
3. Page 4 line 167 ESRS needs to be spelled out.
Thank you this has been updated (page 4, line 169).
4. Was the age different between the children included in the study and those that were not included? If the non-included children were older this would support findings in typically developing children that children may grow out of RMD.
We have run analyses on the age of the 8 index cases at recruitment versus all other children whose parents reported their child rocked at baseline and have found no significant age differences: mean 30.9 months (SD 18.7) versus mean 32.66 months (SD 21.1), p value = 0.7. I have included this in the discussion (page 10, line 294).
5. Table 1 GCSC needs to be spelled out for those readers not familiar with these UK exams. Usually parental education is reported as high school and university undergraduate and postgraduate degrees.
Thank you this has been updated as a footnote under the table (page 6, line 241).
6. Table 2. Suggest removing the 2 decimal places for number of episodes and duration as the data cannot be measured to this level of accuracy.
Thank you for this sensible suggestion. We have now updated the number of episodes, duration of episodes and time spent in RMs columns to reflect this (page 8, line 257).